# Position: To Defend Against Cyber Attacks, We Must Teach AI Agents to Hack

**Terry Yue Zhuo** [1 2]   **Yangruibo Ding** [3]   **Wenbo Guo** [4]   **Ruijie Meng** [5]

## Abstract

For over a decade, cybersecurity has relied on human labor scarcity to limit attackers to high-value targets or generic automated attacks. Building sophisticated exploits requires deep expertise and manual effort, leading defenders to assume adversaries cannot afford tailored attacks at scale. AI agents break this balance by automating vulnerability discovery and exploitation across thousands of targets, needing only small success rates to remain profitable. Current developers focus on preventing misuse through data filtering, safety alignment, and output guardrails. However, such protections fail against adversaries who control open-weight models or develop offensive capabilities independently. We argue that **AI agent-driven cyber attacks are inevitable and require a fundamental shift in defensive strategy**. Defenders must develop offensive security intelligence to predict how attacks will occur at scale. We propose three actions for building frontier offensive AI capabilities responsibly. First, construct comprehensive benchmarks covering the full attack lifecycle. Second, advance from workflow-based to trained agents for discovering in-wild vulnerabilities. Third, implement governance restricting offensive agents to audited cyber ranges and distilling findings into defensive-only agents. Offensive AI capabilities should be treated as essential defensive infrastructure, as containing cybersecurity risks requires mastering them in controlled settings before adversaries do.

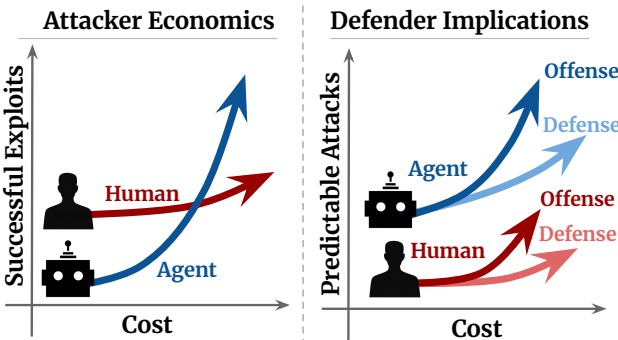

*Figure 1.* Matching AI Attack Scale Requires Autonomous Offensive Security Capabilities. *Left*: AI agents enable economically viable attacks through parallelization. *Right*: Both AI agents and humans can perform offensive or defensive operations, but only offensive AI agents can match the predictability and scale needed to counter AI attackers, and traditional human-scale defense is insufficient.

## 1. Introduction

For more than a decade, software security has depended on continuous human effort, and the shortage of skilled people

[1]Monash University [2]Qwen Team, Alibaba Inc. [3]University of California, Los Angeles [4]University of California, Santa Barbara [5]National University of Singapore. Correspondence to: Terry Yue Zhuo <terry.zhuo@monash.edu>.

*Proceedings of the 43ʳᵈ International Conference on Machine Learning*, Seoul, South Korea. PMLR 306, 2026. Copyright 2026 by the author(s).

has shaped the economic balance for both attackers and defenders (Slayton, 2016). Building a sophisticated exploit often takes deep security expertise and manual work, which motivates attackers to either concentrate on a small number of high-value targets or use automated tools to strike thousands of targets. For example, it took months of effort for attackers to exploit a vulnerability and extract sensitive data from Equifax (Federal Trade Commission). Traditional defenses operate on the assumption of resource asymmetry, where attackers cannot afford to push through multiple layers of protection for every single target (Cashell et al., 2004). In response, the cybersecurity community has long embraced offensive security practices like penetration testing and red teaming, where defenders proactively exploit vulnerabilities in local systems to predict potential attacks and strengthen defenses accordingly (Lynn-Jones, 1995).

**The AI community has not yet developed such a perspective.** Most existing developers focus exclusively on safety-aligned AI and avoid any offensiveness (Bengio et al., 2024), ignoring the potential of offensive security intelligence in the AI era. As AI systems become more powerful and agentic (Wang et al., 2024a), AI agent-driven attacks become inevitable. Recent advances in AI agents could disrupt the traditional attack-defense paradigm, enabling the automation of vulnerability discovery and exploitation at scale (Pot-

ter et al., 2025). Unlike existing automated tools that exploit only known vulnerabilities via pre-programmed rules, AI agents can mimic human strategic agency. Agents adapt to novel systems and discover new attack paths with minimal human guidance, extending economic viability from commodity targets to the long tail of previously ignored systems. Finding and exploiting vulnerabilities resembles software development, and AI agents have demonstrated strong capabilities in software engineering tasks. Recent work suggests AI agents will benefit attackers more than defenders (Potter et al., 2025; Carlini et al., 2025), as current defensive security efforts like vulnerability detection (Chakraborty et al., 2021) are designed to discover vulnerabilities proactively but struggle to predict how attacks will occur. **We argue that defenders must develop offensive security intelligence that proactively exploits vulnerabilities in local systems to predict how attacks might occur.** As shown in Figure 1, offensive security agents allow defenders to model attacker behavior at scale and predict which vulnerabilities are most likely to be exploited under realistic constraints.

The urgency of developing offensive security agents stems from a unique threat profile. First, *AI agents do not need to be expert hackers to be effective attackers*. Cyber attacks already follow an economic logic where attackers are willing to fail repeatedly, and low cost per attempt means only a small number of successes are needed to profit (Laszka et al., 2017; Allodi, 2017). AI agents can automate tasks such as scanning systems, testing vulnerabilities, and continuing attacks after partial success, even if many attempts fail. What matters is not flawless performance but the change in cost and scale. Because AI agents can continuously and adaptively probe thousands of targets at almost no additional cost, the potential return for an attacker increases significantly (Anthropic, 2025).

Second, *existing AI safety mechanisms are easily bypassed by adversaries*. Techniques like filtering training data (Schmitt & Koutroumpis, 2025), safety alignment (Kenton et al., 2021; OpenAI, 2023), and inference-time guardrails (Rebedea et al., 2023) might slow harmful uses of AI temporarily. However, such protections can be broken when attackers control open-weight or self-hosted models. Furthermore, attackers can be AI experts with knowledge of training frontier AI. With the increasing availability of AI infrastructure and decreasing cost of compute resources, attackers can easily remove alignment from AI models and develop offensive AI agents for malicious use, efficiently attacking the long tail of niche or custom systems that were previously ignored due to high manual effort.

The remainder of this paper is organized as follows. We begin by discussing how AI and agentic systems reshape the cyber attack landscape (Section 2), followed by examining the limitations of existing model-centric defenses (Sec-

tion 3). We then outline promising future directions for securing the cyber domain in light of emerging offensive AI capabilities (Section 4). Finally, we present alternative views on the challenges of enabling offensive AI responsibly and scenarios that challenge our foundational assumptions (Section 5). Throughout, our focus is on defenses from the AI perspective rather than system-level defenses that use AI to enhance traditional software security tasks.

## 2. Catastrophic Cybersecurity Risks in the AI Agent Era

In this section, we aim to formalize the frontier cybersecurity risks in the era of AI agents. We believe that AI agents can enable autonomous cyber attacks that break existing software systems. We argue that AI agents introduce not only incremental risks, but also systemic failure modes that can propagate across infrastructure at a pace exceeding human response capacity (XBOW, 2024).

### 2.1. Threat Model

We consider a financially motivated adversary that is technically sophisticated but constrained primarily by human labor, similar to the threat model articulated by Carlini et al. (2025). The adversary has access to state-of-the-art (SOTA) AI agents, either via APIs or local deployment, and can integrate them into automated pipelines for vulnerability discovery, exploitation, and monetization (Fang et al., 2024; Zhu et al., 2024; Jin et al., 2022). We do not assume any nation-state capabilities, novel cryptographic breaks, or privileged access to proprietary infrastructure.

Crucially, the adversary's objective is not to maximize damage to a specific high-value target, but to maximize aggregate profit across a large population of heterogeneous victims. Such objectives can place the threat in the regime where automation and marginal cost reductions have historically driven the most disruptive shifts in attacker behavior. Under this model, failures, hallucinations, or partial exploit success do not meaningfully constrain adversarial effectiveness. Because attacks can be attempted at a massive scale, even low per attempt success rates remain economically viable, particularly as inference costs decline and model access becomes more widespread (Gundlach et al., 2025).

### 2.2. System-Level Vulnerability Exploitation

We note that AI agents can substantially reduce the cost of identifying and exploiting vulnerabilities in the long tail of software systems. Reverse engineering, exploit construction, and validation are among the fixed costs associated with traditional exploit development that are mostly unaffected by the number of users (Maynor, 2011; Allodi, 2017). As a result, attackers concentrate effort on widely deployed

systems where the expected return justifies the investment (Cremonini et al., 2005; Bier, 2007).

Previous empirical evidence suggests that current AI systems are already beginning to lower these economic barriers (Fang et al., 2024; Zhu et al., 2024). AI agents can already autonomously audit small or poorly maintained codebases, identify common vulnerability patterns, and generate actionable bug reports with limited human oversight (Xu et al., 2024). While such vulnerabilities are often low sophistication, their prevalence across undersecured systems creates a vast, previously uneconomical attack surface. The system-level implications of automated vulnerability discovery are particularly concerning. Some of the existing infrastructure systems, particularly, rely on barely maintained components, such as embedded systems and niche web services. The exploitation of nearly deprecated components will provide initial access vectors that are less noticeable. We foresee that once attackers gain initial access through the weakly secured entry points, they may move into more sensitive parts of the entire system. The result is cross-domain compromise chains that connect seemingly isolated security failures into paths of escalating access and damage.

### 2.3. Automated Superhuman Cyber Attacks

Beyond vulnerability discovery, AI agents enable a class of attacks characterized by adaptive post-exploitation behavior. Once code execution or authenticated access is obtained, AI agents can analyze the compromised environment, identify high-value assets, and tailor their actions to the specific victim. The capability of AI agents will reduce the marginal cost of customized attacks. Historically, attackers have relied on generic monetization approaches, such as ransomware, as per-victim customization was prohibitively expensive (Cremonini et al., 2005). AI agents may invert such calculus by reasoning over tons of documents, images, audio, and system state, enabling attacks that extract maximal value from each compromised system while remaining scalable.

The convergence of breadth and depth represents a qualitative shift in the cyber threat landscape. Defensive mechanisms, such as static signatures and log monitoring, are optimized for well-known attacks (Iyer, 2021). In contrast, AI agents can vary their attack methods significantly, adjust their strategies when they encounter security measures, and operate within legitimate system functionality, including navigating interfaces and performing actions like those of authenticated users. At scale, automated systems create the conditions for superhuman cyber attacks. AI agents may not routinely discover new zero-day vulnerabilities, but they can perform entire attacks faster and more effectively than human attackers (CrowdStrike, 2025). When combined with declining inference costs and increasing autonomy, AI

agent-driven attacks make widespread security breaches more likely. Once attackers compromise one system, they can more easily spread to other connected software systems and networks.

## 3. Defensive Safeguarding Against Cyber Attackers is Not Enough

To reduce the possibility that AI agents will be abused for cyberattacks, a variety of defensive measures have been put forth and implemented. The majority of protections rely on preset guidelines and limitations that are implemented at particular stages of the AI system. Instead of creating intelligent defensive systems that can actively thwart attacks, developers try to limit model behavior through localized controls on training data, model outputs, or user access. Although the aforementioned defensive strategies offer certain protection against abuse, they all depend on presumptions about the resources and capabilities of attackers. The beliefs that underpin existing protections become less trustworthy as AI agents grow more sophisticated and widely available. In this section, we examine the major categories of defensive safeguards and explain why each approach falls short against adaptive adversaries using AI agents.

### 3.1. Data Governance

**What It Offers** Data governance aims to reduce cyber risk by controlling the data that AI systems can process and the information they can learn from. Common practices include filtering pre-training corpora to remove vulnerable code snippets, malware, leaked credentials, and other clearly harmful artifacts. Dataset auditing, deduplication, and documentation are also common (Bengio et al., 2024). Additional mechanisms are implemented to restrict access to private or regulated data through PII detection, redaction, and privacy-preserving training methodologies (Feretzakis et al., 2024). Some developers will check user inputs for harmful or private information at inference time and stop logging or keeping such data that may be used to train the model (Dainotti et al., 2012).

**How It Fails** Data governance does not address the primary source of cyber risk introduced by AI, like general-purpose reasoning and code synthesis. Many cyber attacks do not depend on memorized exploit text, but instead arise from the ability to reason about systems, infer vulnerabilities, and construct novel attack logic from the first principles (Zhu et al., 2024). Removing explicit exploit examples does not eliminate these capabilities, particularly for the long tail of software systems that never appeared in the training data (Carlini et al., 2025). As AI becomes more agentic and capable of tool use, it can also acquire new information at inference time, further weakening the connection between

training data controls and downstream behavior. Data governance assumes that harmful behavior can be traced to specific data sources (Janssen et al., 2020), while attackers benefit from probabilistic success and defenders must filter comprehensively, creating an inherent asymmetry.

## 3.2. Safety Alignment

**What It Offers**   Safety alignment techniques attempt to constrain model behavior through supervised fine-tuning, reinforcement learning, and preference optimization (Mu et al., 2024; OpenAI, 2023). It is suggested that alignment will help discourage harmful actions and induce refusals for disallowed requests. Alignment is often combined with red teaming and post-training evaluations intended to surface obvious misuse cases prior to deployment (Ji et al., 2025).

**How It Fails**   The safety alignment is inadequate to resist attackers. There are no restrictions on the kind of cues or actions with which an AI agent can be effectively aligned, and methods for jailbreaking are continuously evolving (Andriushchenko et al., 2025). We argue that objective distortion can be enough to bypass restrictions. Furthermore, we note that harmful behavior may emerge from the composition of individually benign steps, particularly in long-horizon trajectories where AI agents plan and interact with external tools. Some work shows that alignment mechanisms optimized for single-turn conversations still struggle to detect malicious behaviors in the long context (Lynch et al., 2025). Alignment also degrades under fine-tuning or retraining, which attackers can perform at low cost once models are accessible (Qi et al., 2024).

## 3.3. Representation Engineering

**What It Offers**   Representation engineering strategies seek to alter internal model representations to either suppress or enhance particular behaviors (Mitchell et al., 2022). Prior studies have explored feature steering, activation editing, and intervention to keep hazards in check without affecting overall performance (Zou et al., 2023; Ghandeharioun et al., 2024; Wang et al., 2025a). These approaches provide the potential for more precise control than prompt design and can be implemented without changing the training data.

**How It Fails**   Controls at the representational level often fail when applied to unseen scenarios. We argue that small changes to the given context can bypass the engineering effort. Internal representations are complex and heavily depend on what the model is doing (Tan et al., 2024; Zhang et al., 2024). When AI acts as agents, their behavior emerges from extended interactions rather than a single internal state, making specific edits to representations less effective (Wehner et al., 2025). Fully verifying representation engineering methods remains difficult, as they often do not provide clear guarantees about model behavior in new or adversarial contexts (Tan et al., 2024).

## 3.4. Output Guardrails

**What It Offers**   Output guardrails operate at inference time and attempt to detect or block harmful content. Common approaches include prompt classification, output filtering, moderation models, and rule-based checks applied before responses are returned to users (Ayyamperumal & Ge, 2024). Guardrails are attractive because they can be updated independently of model training and deployed selectively across applications (DONG et al., 2024).

**How It Fails**   Guardrails work on the assumption that harmful intent or behavior shows up in individual prompts or responses. However, in agentic workflows, harmful outcomes can arise from sequences of seemingly harmless actions (Gu et al., 2024; Kumar et al., 2024). We also note that guardrails will be effective when AI is deployed as open-weight, self-hosted, or embedded systems where monitoring is limited or absent. Attackers can slip past detection by hiding their intentions, using indirect methods, or dispersing malicious actions across multiple steps through tool-based interactions (Jin et al., 2024; Villa et al., 2025).

## 3.5. Access and Deployment Controls

**What It Offers**   Access controls and deployment restrictions try to limit misuse by regulating who can use models and under what conditions. UK AI Safety Institute (2025) and Eiras et al. (2024) show that AI organizations that require API-only access and licensing terms, generally release more powerful models. Centralized deployments let developers track how models are being used, cut off access when needed, and enforce policies.

**How It Fails**   We argue that access controls fall apart once models become widely available. Open weight releases remove all serving-time restrictions, and even gated models can leak, get replicated, or be independently reimplemented (Rigaki & Garcia, 2023; Liang et al., 2024). As discussed in Section 3.2, fine-tuning on a small dataset can break the safety-aligned behaviors. We consider deployment controls static, since there is no effective way to update safeguards of the open-weighted models. As capabilities spread across the ecosystem, restricting individual deployments does little to prevent misuse at the system level (Bengio et al., 2024).

*Table 1.* Performance of SOTA agents on widely used security benchmarks. The numbers are higher, the better.

| Capability | Dataset | Performance (%) |
|---|---|---|
| Attack generation | CyberSecEval-3 (Wan et al., 2024) | 49% |
| | SeCodePLT (Nie et al., 2025b) | 0.2% |
| | AutoPenBench (Gioacchini et al., 2024) | 54.5% |
| | CVE-bench (Zhu et al., 2025b) | 12.5% |
| CTF | CyBench (Zhang et al., 2025b) | 55% |
| | NYU (Shao et al., 2024) | 22% |
| Vul. detection | PrimeVul (Ding et al., 2024) | 12.9% |
| | VulnLLM (Nie et al., 2025a) | 77.8% |
| PoC generation | CyberGym (Wang et al., 2025b) | 28.9% |
| Patching | SEC-bench (Lee et al., 2025) | 22.3% |
| | SWE-bench-Verified (Yang et al., 2024) | 78.8% |

# 4. Future Safeguarding with Offensive Security Agents

## 4.1. Frontier Offensive Security Measurements

Properly and comprehensively measuring the offensive security capabilities of AI and AI agents is the essential first step towards understanding their potential risks and improving their defense capabilities. This requires constructing high-quality benchmarks that cover comprehensive cyber attack steps and defense pipelines. Existing works have constructed a set of benchmarks, where most of them are developed for standalone models. Such benchmarks only provide static datasets without dynamic evaluation environments. More recent works start to explore more realistic agentic-facing benchmarks, where they provide the whole software projects as well as the dynamic execution environment (e.g., Docker files) and proper metrics.

Table 1 summarizes the SOTA cybersecurity benchmarks, including both attacks and defenses, as well as the best performance on these benchmarks. First, the table shows that the current benchmarks still do not cover the full attack and defense lifecycle, and they are not fine-grained enough. For example, certain critical attack steps, such as exploit chaining and command and control, are not covered (MITRE, 2024; Martin, 2016). On the defense side, existing benchmarks do not cover project-level vulnerability detection and root cause analysis. Besides, the patching benchmarks also have limited vulnerability type coverage as well as limited benign testing cases.

Second, the SOTA agents' performances on different capabilities vary a lot. At a high level, today's AI agents *perform better on small-scale generative tasks than large-scale analytic tasks*. For example, the performance on patching short functions (SWE-bench) is better than PoC generation on large projects (CyberGym). Here, the PoC generation task gives the agent a whole project without any label of which functions are vulnerable. This requires an agent to analyze the whole project, understand the data and control flow, as well as project semantics. The agent also needs to have a good understanding of the security principles, identify

potential vulnerable locations, and resolve complex branch conditions to generate vulnerable inputs that can reach the target location and trigger the vulnerabilities. On the attack side, exploiting chaining is a much more difficult task than reconnaissance (e.g., writing and sending phishing emails).

Looking forward, it is important to develop more comprehensive cybersecurity benchmarks that cover fine-grained attack and defense categories, include large-scale and real-world projects, and provide dynamic execution environments and proper metrics. We can build new benchmarks based on the attack lifecycle specified in MITRE (MITRE, 2024) and the cyber kill chain (Martin, 2016). Different real-world attacks may target different steps. To make sure realism and coverage of attacks, we can collect real-world attacks focusing on different steps and distill attack playbooks for these attack steps. Then, we can construct simulated systems covering all necessary components of collected attack books and construct attack tasks based on the playbook. We can create multiple system variations to cover different types of systems. Given that systems are created and maintained by benchmark constructors, it is also easy to provide corresponding dynamic execution environments. To make the benchmark even more agent-friendly, it is also critical to provide proper tool sets as well as agent scaffolds. Here, providing security-specific tools, such as static and dynamic program analysis tools, rather than solely the general bash tools, would be more helpful to evaluate the agents' cybersecurity-specific capabilities.

Benchmark quality control is also critical. First, we need to make sure the environment is robust and does not contain apparent flaws. Recent research shows that for CTF benchmarks, if the environment contains flaws, the agent may take shortcuts to exploit the environment flaws rather than solving the designed tasks (Meng et al., 2025). Second, the ground truth label or judge needs to be correct. This is especially important for vulnerability detection, as most of the existing vulnerability detection labels are noisy due to multiple reasons (e.g., lack of the necessary context) (Ding et al., 2024). Finally, in light of the rapidly evolving dynamics between attackers and defenders, cybersecurity benchmarks must be updated on a regular basis to remain aligned with the most recent attack techniques and threat landscapes.

## 4.2. Frontier Offensive Security Development

Understanding how AI may be used for offensive security is increasingly important for two reasons. First, it helps anticipate how future attackers might operate once autonomous AI agents become widely accessible (Wallace et al., 2025). Second, it enables defenders to proactively identify and fix vulnerabilities before those capabilities are misused in the wild (Wang et al., 2025b). Offensive security intelligence is therefore no longer limited to modeling human adversaries.

*Table 2.* Practical evolution path of offensive security agents.

| Stage | Capabilities | Development Approaches | Current Status |
|---|---|---|---|
| Knowledge models | Security issue analysis | Domain-specific pre-training | Mature and widely available |
| Workflow agents | Vulnerabilities exploitation from limited scenarios | Prompting with external orchestration | Finding non-critical vulnerabilities and CTFs |
| Trained agents | Zero-day vulnerability discovery | Post-training from cybersecurity environments | Project Glasswing (Anthropic, 2026) and Trusted Access for Cyber (OpenAI, 2026) |

It must also account for machine-driven ones and explore how similar capabilities can be leveraged for defense.

*The development of AI agents in other domains (e.g., software engineering) can serve as a reference for where offensive security agents are likely heading.* Early systems relied heavily on external workflows that combined large language models with retrieval, patch generation modules, test execution, and repair loops to fix bugs in codebases (Yang et al., 2024; Wang et al., 2024b). AI agents work well because the expert-designed workflow will help AI behave like human practitioners. As runtime environments became available and verifiable, researchers began training models to learn repair behaviors instead of depending on fixed pipelines (Pan et al.; Wei et al., 2025).

Offensive security agents today largely remain in the workflow stage, where agents access Ghidra decompilers, network scanners, and exploit frameworks through external tool calls (Deng et al., 2024; Abramovich et al.). Agents can perform penetration-style tasks through scaffolding, although their ability is limited by fixed pipelines and brittle reasoning. Systems do not accumulate experience or improve their strategies over time.

We suggest that cybersecurity offers a strong opportunity to move beyond the workflow stage, as stated in Table 2. For instance, the community is actively constructing cyber ranges and controlled environments where vulnerabilities and exploitation mechanisms are well understood (Ferguson et al., 2014; Yamin et al., 2020). Such cyber ranges produce detailed records of how exploits unfold, from reconnaissance through analysis, exploit construction, and successful compromise. It is quite straightforward to convert them into runtime environments for high-quality trajectory collection. Beyond that, cyber ranges are well-suited for reinforcement learning because outcomes are directly measurable. An exploit either succeeds in triggering a vulnerability or fails (Zhu et al., 2025b; Wang et al., 2025b), encouraging AI to discover strategies that go beyond what humans previously document.

Given the precedent from software engineering and the availability of verifiable cyber environments, we expect offensive security agents to evolve from workflow-driven systems to trained and adaptive ones. The direction creates an im-

portant defensive advantage. As AI agents improve in autonomously discovering vulnerabilities, defenders gain the same capabilities to accelerate patching and reduce the window between vulnerability emergence and remediation. We believe that advancing offensive AI responsibly depends on strengthening long-term defensive preparedness in parallel.

### 4.3. Frontier Offensiveness Protection

Teaching AI agents to hack is only defensible, which is our ultimate goal, if we can ensure that the offensive capability will be used as defensive instrumentation. The ideal outcome should be a diagnostic adversary used to surface vulnerabilities before deployment, shorten the discovery-to-remediation loop, and continuously verify resilience under evolving attack strategies. To realize offensive protection, the central challenge is how to properly govern, contain, and translate offensive capability into deployable defense. We note that the governance problem considered here differs from most existing AI safety frameworks. Current governance efforts primarily focus on restricting the deployment or proliferation of dangerous frontier capabilities through access controls, release policies, and risk management frameworks (?UK AI Safety Institute, 2025). In contrast, our setting assumes that offensive AI capabilities will continue to emerge and asks how such capabilities can be responsibly developed, contained, audited, and operationalized for defensive purposes within controlled environments. The central challenge is therefore not only capability restriction, but also how to safely translate offensive intelligence into deployable defensive utility without enabling uncontrolled misuse or leakage. We outline a protection framework for frontier offensiveness with three mechanisms: (1) version control and staged release of offensive capabilities, (2) offensive agents' restriction to audited, controlled environments, and (3) using offensive agents to train or supervise defensive-only agents that can be safely released. Together, these mechanisms make offensive capability practically useful for pre-release security while minimizing misuse and leakage.

First, offensive agent capability should be treated as a high-risk artifact whose distribution is gated by measured competence. We advocate for managing the offensive capa-

bility through *capability-tiered* checkpoints, where each model is versioned, evaluated on standardized offensive measurements (Section 4.1), and assigned to a release tier that determines where and how it may operate. The proposed staged-release regime aims to ensure that different levels of offensiveness can be separately established so that we can always use the strongest adversary in a timely but possibly regressive manner, while preventing the increasingly autonomous exploitation competence from causing real damage in open settings where existing safeguards are brittle.

Second, the primary protection boundary for an offensive agent must be realized as an audited, controlled environment. To prevent leakage or unintended harm, organizations must dedicate effort and resources to building and maintaining strictly audited cyber ranges. The construction of these environments is a foundational security requirement, necessitating faithful replicas of real systems that are designed with strict network isolation and tool access controls to prevent unintended interactions with the public internet, ensuring that offensive capabilities remain contained within a sandbox of least privilege. With this safe-by-design environment, offensive security capability functions as defensive instrumentation, where the ultimate deliverable results are the defensive improvements it induces, such as comprehensive, tamper-evident logs of every tool call and failure, which allows defenders to learn from attack trajectories and come up with actionable defensive heuristics.

To further reduce the risk of misuse, offensive capability should remain under human-in-the-loop governance throughout the offense-to-defense pipeline. Attack trajectories generated inside cyber ranges should not be directly transferable to open deployment environments before defensive artifacts have been produced and validated. Instead, security researchers should review exploit traces, validate the discovered vulnerabilities, and supervise the translation of offensive findings into defensive outputs such as patches, regression tests, detection signatures, and remediation strategies. During this process, offensive trajectories remain restricted within the audited cyber range, reducing the dangerous window between attack discovery and defensive response. The goal is not to operationalize autonomous offensive capability in real-world systems, but to use controlled offensive intelligence to accelerate secure remediation before similar attack strategies emerge in uncontrolled settings.

Finally, to maximize defensive benefit while minimizing proliferation risk, organizations should decouple those models trained for offensive discovery from the defensive deployment of secure agents. The ultimate goal of teaching agents to hack is to produce superior defensive systems that can operate at machine speed without inheriting the danger-

ous action space of an attacker. In this offense-to-defense workflow, an offensive agent identifies and validates vulnerabilities in containment, and these traces are then distilled into actionable security artifacts, such as automated patch suggestions and regression tests. While the offensive agent remains restricted to the cyber range, specialized defensive agents, which focus exclusively on detection, root cause analysis, and remediation, can be safely released to protect the global software ecosystem. This separation ensures that the loop between discovery and repair is closed by machine-scale intelligence, allowing defenders to secure the rare or unknown cases of software that were exploited by offenders but currently receive limited adversarial attention.

## 5. Alternative Views

Our argument that securing the future requires investment in offensive intelligence rests on assumptions about AI capabilities, attacker adoption, and the limits of existing safeguards. In this section, we consider alternative views that challenge these assumptions and explain why they do not eliminate the need for our approach.

### 5.1. Challenges in Teaching AI Agents to Hack

First, training agents as good or even expert hackers can be more challenging than other agentic capabilities due to the uniqueness of cybersecurity. The challenges come from the following aspects. First, the attack data is not easy to obtain, although standard databases provide vulnerabilities (e.g., CVE database), certain complex real-world attacks against real-world systems are hard to obtain, as releasing them may raise ethical issues. Besides, even for relatively easy-to-obtain data, the data quality is hard to guarantee. For example, labeling malware and vulnerabilities is a time-consuming process that requires extensive expertise compared to labeling images. Second, solving security tasks (both attacks and defenses) requires using domain-specific tools that most existing agents have not learnt yet. When it comes to code-related tasks, existing agents still tend to call common bash and search tools. AI and agents still have limited understanding and capabilities against domain-specific tools, such as kali, CodeQL, and fuzzers. Teaching AI to use these tools requires new system environments, agent scaffolds, and new learning algorithms. Finally, it is common to have long-tail and out-of-distribution (OOD) tasks in security. Attack evolution may introduce distribution shifts that existing AI models cannot handle. Although large AI models reduce OOD issues, it is still challenging to train an agent that consistently performs well. Solving such challenges requires deep collaboration between both the ML and the security community. Even if the technology is developed to a point where all the challenges mentioned above can be tackled, and the agents achieve the capabilities

of expert hackers. It then becomes even more challenging to ensure that such capabilities are only used by responsible security researchers (white-box hackers), not real attackers. Given the general trade-off between security and utility, relying solely on model safety alignment may be challenging. One possible solution is to enforce system-level access control or privilege isolation to ensure only authorized users can access the deep attack capabilities. Overall, ensuring the responsible use of frontier AI capabilities in offensive security is a significant challenge that both AI and systems researchers must address.

### 5.2. Risks of Offensive AI Falling into the Wrong Hands

Another alternative view is that developing frontier offensive AI capabilities may ultimately decrease security rather than improve it, because once such systems are created they may inevitably leak, be stolen, replicated, or commercialized in ways that empower malicious actors. This perspective argues that investing in offensive intelligence risks providing effectively free research and development for attackers, lowering the cost of sophisticated cyber operations and accelerating the spread of advanced attack capabilities. The concern is particularly credible because cybersecurity knowledge has historically diffused rapidly once demonstrated, and offensive techniques developed by governments or elite security teams have often later appeared in criminal ecosystems. As AI agents become increasingly capable of automating vulnerability discovery, exploitation, and post exploitation tasks, the consequences of capability leakage may become substantially more severe. From this perspective, intentionally advancing offensive AI may unintentionally strengthen adversaries faster than it improves defense. However, avoiding the development of offensive AI capabilities does not eliminate this risk. The relevant strategic question is not whether offensive AI will exist only in responsible hands versus not exist at all, but whether defenders can understand and shape these capabilities before attackers develop them independently. The incentives for malicious actors to pursue automated offensive intelligence remain strong as frontier models become more widely accessible and AI capabilities continue to improve. If responsible security researchers avoid this area entirely, attackers may still develop offensive agents without corresponding defensive preparation, governance mechanisms, or safety infrastructure. Moreover, research on controlled access, monitoring, privilege isolation, and secure deployment depends on direct engagement with offensive capabilities themselves. Although the risks of misuse and capability diffusion are substantial, refusing to study offensive AI may ultimately leave defenders less prepared for a future in which such capabilities emerge regardless.

### 5.3. Limited Adoption of AI Agents in Cyberattacks

One alternative view is that AI agents will see limited real-world adoption by cyber attackers. This perspective holds that while AI demonstrates impressive capabilities in controlled experiments, deploying them reliably in adversarial, noisy, and high-risk environments remains difficult. Cyber attacks often require robustness, stealth, and operational discipline, and attackers may be reluctant to rely on systems that are probabilistic, costly, or prone to failure (Rid & Buchanan, 2015). From this viewpoint, AI-based attacks may remain niche tools rather than becoming a dominant force, reducing the urgency of developing frontier offensive intelligence. However, historical patterns suggest that once automation meaningfully lowers costs, adoption tends to follow rapidly, even when tools are imperfect (Kaloudi & Li, 2020). Many successful attack techniques, from phishing kits to exploit frameworks, were initially unreliable yet still proved economically viable at scale. Moreover, AI agents need not replace human attackers entirely to be transformative. Even partial automation of vulnerability discovery, reconnaissance, or post exploitation tasks can significantly shift attacker economics. As AI capabilities improve and inference costs decline, the barrier to adoption is likely to decrease, making limited uptake an unstable equilibrium.

## 6. Related Works

**Large Language Models for Cybersecurity**   Research on large language models for cybersecurity has progressed from early domain-adaptive encoder models to scalable generative architectures enabled by curated security corpora. Early models such as CyBERT (Ranade et al., 2021), Secure-BERT (Aghaei et al., 2022), and CTI-BERT (Park & You, 2023) demonstrated the benefits of domain-specific fine-tuning, but closed datasets and task-specific adaptation limited scalability. More recent work emphasizes data-centric approaches based on continued pretraining and instruction tuning. PRIMUS (Yu et al., 2025) and Foundation-Sec-8B (Kassianik et al., 2025) are pretrained on large-scale cybersecurity corpora and then adapted via post-training strategies, though their datasets remain unreleased. CyberPal (Levi et al., 2025a) introduces expert-driven cybersecurity instruction tuning to improve reasoning and instruction following, while CyberPal 2.0 (Levi et al., 2025b) further extends this approach by training smaller specialized modes using enriched expert-curated data.

**Agentic Defensive Security**   Recent progress in agentic defensive security has explored AI agents that leverage program analysis, reasoning, and iterative planning to enhance traditional vulnerability discovery and mitigation processes. RepoAudit introduces an autonomous LLM agent for repository-level code auditing that navigates large code-

bases, performs on-demand analysis, and incorporates validation to reduce false positives (Guo et al.). VulnLLM-R presents a specialized reasoning LLM with an agent scaffold designed to detect vulnerabilities by reasoning about program state, outperforming both static analysis tools and general reasoning models (Nie et al., 2025a). Beyond detection, fuzzing and exploit generation have also adopted AI agents. Locus implements an agentic framework for synthesizing semantically meaningful predicates to guide directed fuzzing, substantially improving efficiency and uncovering previously unpatched bugs (Zhu et al., 2025a). Similarly, PBFuzz automates the expert workflow for proof-of-vulnerability input generation by iteratively extracting reachability and triggering constraints, synthesizing test strategies, and leveraging feedback to satisfy complex constraints (Zeng et al., 2025).

**Agentic Offensive Security**    Agentic offensive security explores the use of AI agents to perform multi-step penetration testing, vulnerability exploitation, and Capture The Flag (CTF) tasks in interactive environments. Early systems such as PentestGPT (Deng et al., 2024) illustrate the feasibility of applying LLMs to offensive workflows, though they rely heavily on human guidance. More recent approaches focus on higher degrees of autonomy through structured agent design and environment interaction. EnIGMA (Abramovich et al.) introduces an agentic framework tailored for CTF challenges, integrating tool execution, iterative reasoning, and feedback-driven planning to solve complex offensive tasks end to end. Recent works (Zhuo et al., a;b) have started to address the scarcity of long-horizon training data by synthesizing interaction trajectories for offensive agents, enabling improved performance and generalization across multiple CTF benchmarks. Progress in this area is further supported by the development of standardized evaluation environments and benchmarks, including Cybench (Zhang et al., 2025b), CVE-Bench (Zhu et al., 2025b), and Bounty-Bench (Zhang et al., 2025a), which assess agentic capabilities across professional CTF tasks, real-world vulnerability exploitation, and impact-driven bounty scenarios.

## 7. Conclusion

In this work, we argue that the current defensive AI safety paradigm poses a restrictive view of cybersecurity resilience in the age of AI agents. Focusing solely on model-centric safeguards remains disconnected from the economic reality that AI agents fundamentally alter the cost structure of cyber attacks. We posit that developing offensive security intelligence should be recognized as essential defensive infrastructure, as relying exclusively on reactive protections continues to widen the gap between what attackers can automate and what defenders can anticipate. Only by proactively teaching AI agents to hack within controlled environments can

we model adversarial behavior at scale, predict exploitation patterns before they materialize, and maintain a defensible security posture rather than perpetually responding to threats we cannot foresee. The choice is not whether offensive AI capabilities will exist, but whether defenders will master them under responsible governance or be forced to reverse-engineer them after attacks have already succeeded.

## Impact Statement

This paper examines the dual-use cybersecurity implications of increasingly capable AI and argues for the development of defensive intelligence informed by controlled offensive research. As AI becomes more agentic and integrated into critical digital infrastructure, failures to anticipate and mitigate its misuse could lead to large-scale security, privacy, and economic harms. While research into offensive capabilities raises ethical concerns around misuse and leakage, avoiding such study may leave defenders unprepared for adversaries who explore these capabilities independently. We emphasize that offensive intelligence should be developed only within secure, well-governed research environments and used to strengthen defensive systems rather than enable real-world attacks. Overall, this work aims to support the responsible advancement of machine learning by improving the resilience of digital systems to emerging automated threats.

## Acknowledgement

We thank Peng Chen, Xiaolong Jin, and Haoran Lyu for their comments.

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
