# OpenReview forum: "Position: To Defend Against Cyber Attacks, We Must Teach AI Agents to Hack"
_ICML.cc/2026/Position_Paper_Track — ICML 2026 Position Paper Track regular_

### Official Review · Reviewer_j1aM · 2026-02-23

**Significance:** 2
**Argument Clarity:** 3
**Rating:** 4
**Confidence:** 2

**Questions:**

- Can the authors address the points stated in **W1** and **W2**
- Moreover, could they add a discussion about the difference of their governance related suggestions to existing works in this domain? (**W3**)

**Alternative Views Section:**

Yes

**Compliance With Llm Reviewing Policy A Conservative:**

Affirmed.

**Discussion Potential:**

2

**Final Justification:**

Changed score based on the author response and the discussion with other reviewers

**Paper Summary:**

The authors argue that the cost of conducting cybersecurity attacks is decreasing significantly in the age of LLMs. This is backed up by literature and industry references. Given this observation, they argue that we need to build AI pen-testing systems to assess emerging vulnerabilities posed by agentic attackers. Their main suggestions are: 1) building improved benchmarks, 2) relying more on agentic models to uncover vulnerabilities, 3) training governance restricted offensive agents and transfering the learning that they bring regarding vulnerabilities to defensive models.

**Position:**

Yes

**Position In Title:**

Yes

**Related Work:**

3

**Strengths And Weaknesses:**

**Major weaknesses**

**W1:** While I agree with the position that the cybersecurity arms race is fundamentally shifting, I at the same time believe that the authors basically state a majority opinion that is largely already being implemented by both academia and industry. Thus, I do not see a considerable effect of this position paper on the research community, and for example, provoke an interesting discussion. The authors themselves provide numerous references, where both academia and industry identify agents as a considerable cybersecurity threat. My arguments are given in the next comments:

1) Companies and tools such as XBOW, HexStrike AI, AISLE, Aikido, Pentera, and recently also Anthropic, to some degree, all provide agent-driven pen-testing services to uncover and fix vulnerabilities. They seem to have already implemented the authors' second and third suggestions to some degree (it is impossible to say exactly how they address these problems, but they use pen testing and combine it with vulnerability fixes).

2) Academia is also rapidly picking up on the importance of agents in cybersecurity, providing new benchmarks and showcasing new attack surfaces (as referenced by the authors). Benchmarks are always an active research topic (specifically in LLMs) and I do not see a novel position here.

**W2** I find the positions of the counterargument to be relatively weak. They are mostly concerned with the feasibility of implementing the proposed solution. However, this feasibility has already been demonstrated by successful public efforts to close security vulnerabilities involving AI agents. I would have liked more arguments in the direction of 5.3 that focus on other aspects of the position. For example, insofar as it is already being implemented / what aspects exactly are missing from current industry and academic efforts / if this is even relevant, or things are already moving in the right direction / etc.

**Minor weaknesses**

**W3** Governance-related arguments are not backed up by literature in this direction. (e.g., 4.3. Frontier Offensiveness Protection). There is a lot of work on AI governance (International AI safety report, academic work, etc.). These make similar suggestions and it would be nice to know how the suggestions of the authors deviate from these documents.

**W4** Remove placeholder appendix

**Support:**

2

---

> ### Author Rebuttal · Authors · 2026-03-30
>
> > W1: The authors basically state a majority opinion that is largely already being implemented by both academia and industry.
>
> There is a fundamental distinction between what existing companies are doing and what we advocate for. Companies such as XBOW, HexStrike AI, AISLE, Aikido, and Pentera are all operating at the workflow agent stage, as explicitly captured in Table 2, where agents rely on prompting with external orchestration and fixed pipelines to find non-critical vulnerabilities. These systems do not accumulate experience, do not improve their strategies over time, and are not trained from cybersecurity environments. What we advocate for, namely trained offensive agents that use post-training from cybersecurity environments to discover zero-day vulnerabilities autonomously, is explicitly marked as underexplored in Table 2 and is fundamentally different from what these companies currently offer. The reviewer's claim that these companies have already implemented our second and third suggestions therefore reflects a misunderstanding of the distinction between workflow agents and trained agents that is central to our argument.
>
> Regarding Anthropic specifically, their shift toward dual-use cybersecurity capabilities is very recent, occurring only in late 2025, and actually reinforces rather than undermines our position. The fact that even the most prominent AI safety organization has only just begun making this shift demonstrates that the cultural and institutional barriers we identify are real and only now beginning to break down, which is precisely why this position paper is needed at this moment.
>
> Regarding academia providing new benchmarks, our first suggestion goes significantly beyond what existing benchmarks provide. As we discuss in Section 4.1 and Table 1, existing benchmarks still do not cover the full attack and defense lifecycle, lack fine-grained coverage of critical attack steps such as exploit chaining and command and control, and do not provide project-level vulnerability detection and root cause analysis on the defensive side. Our position is not simply that benchmarks should be built, which is indeed well accepted, but that the specific type of comprehensive, lifecycle-covering, dynamically executable benchmarks we describe do not yet exist and need to be built.
>
> > W2: The counterarguments are relatively weak and mostly concerned with feasibility.
>
> We fully agree that the alternative views section could be significantly strengthened and will revise it in two ways. First, we will demote the "stalled AI progress" scenario from a standalone section to a brief acknowledgment, as it represents a relatively weak counterargument. Second, and more importantly, we will add a substantive new alternative view addressing the risk that offensive AI capabilities may fall into or be sold to the wrong hands, which represents the most credible counterargument to our position. We will also add a discussion explicitly addressing what aspects are currently missing from existing industry and academic efforts and why the current trajectory, while moving in the right direction, is insufficient without the principled framework we propose.
>
> > W3: Governance-related arguments are not backed up by literature.
>
> We respectfully disagree and believe there is a misunderstanding of what Section 4.3 proposes. Our governance framework is not concerned with general AI safety governance but with a specific and novel problem: how to responsibly develop, contain, and operationalize offensive AI capabilities within audited cyber ranges without leakage, and how to translate findings into defensive artifacts. This is a fundamentally different governance problem from what existing frameworks address. The International AI Safety Report and the UK AI Safety Institute's work on open-weight model risk management are primarily concerned with preventing AI from being developed or used for catastrophic risks. Their position is essentially the opposite of ours: they aim to restrict and avoid offensive AI capabilities entirely. Our governance framework, by contrast, accepts that offensive AI capabilities must be developed and asks how this can be done responsibly within controlled environments. This represents a genuinely novel governance challenge that existing literature does not address, precisely because the dominant paradigm in AI governance has been avoidance rather than responsible development and containment of offensive capabilities. We will make this distinction more explicit in the revised version by directly engaging with existing governance literature and clearly articulating how our framework departs from and extends the current consensus.
>
> > W4: Remove placeholder appendix.
>
> We thank the reviewer for catching this. The placeholder appendix was included by mistake and will be removed in the final version.

---

> > ### Author Rebuttal · Reviewer_j1aM · 2026-03-31
> >
> > Thank you for the response,
> >
> > Some concerns have been addressed:
> > - Strengthening of the alternative views
> > - Governance related arguments
> >
> > I have to say that the remaining concerns mostly remain.
> > - This is a very prominent topic. The suggestions made by the authors are to high-level to justify acceptance just because of them.
> >
> > However, I read the other reviews and author response and would like to note that position papers will by nature be less objective to judge as other works and I seem to reflect a minority opinion. I adjusted my score to reflect that I am not against the acceptance of this paper in principle and see that it could benefit some people in the community.

---

### Official Review · Reviewer_fNn4 · 2026-03-02

**Significance:** 3
**Argument Clarity:** 3
**Rating:** 5
**Confidence:** 4

**Questions:**

- In what timeframe do you expect your proposed methods to be effective?
- Is there anything stopping this development now other than capability? Is there someone who needs to read this and change their behavior?
- What would the process look like to turn AI-discovered attacks into defenses? Do you anticipate a period of time where the attacks exist in multitudes and the defenses have not yet been created?

**Alternative Views Section:**

Yes

**Compliance With Llm Reviewing Policy A Conservative:**

Affirmed.

**Discussion Potential:**

2

**Final Justification:**

The rebuttal reinforced my prior assessment. I recommend acceptance.

**Paper Summary:**

The paper is concerned with computer security. It argues that the growth of agentic AI makes it inevitable that increasingly sophisticated cyberattacks will be created at a decreasing cost. This allows for cyberattacks at scale, in a way that is not throttled by attacker expertise, time, or resources. Human defenders of computer networks will be unable to keep up with this growth and scale, meaning that AI-automated red teaming must be considered an essential part of computer security work.

The paper lays out the AI-facilitated attack vector, describes why techniques to limit AI systems' ability to create cyberattacks will not be effective, describes what would be necessary to make an AI red-teaming system effective, and collects automated red teaming examples as related work.

**Position:**

Yes

**Position In Title:**

Yes

**Related Work:**

2

**Strengths And Weaknesses:**

**Strengths**

- The argument is clearly stated and easy to follow. The logical flow is good, and makes a strong argument.
- The topic is important and well-timed - it relates to capabilities that largely do not yet exist, but will likely exist soon, making its timing appropriate.
- Related Works covers quite a bit of recent citations in the field.

**Weaknesses**

- It is not clear that this is a position that needs to be argued. Is there reason to believe the cybersecurity community's belief in red teams does not extend to AI-generated attacks? Or is the audience the AI community who is *not* from the cybersecurity community (which may, admittedly, have a different view)? I am not convinced this will lead to significant discussion as the ethics and utility of red teaming in other ways strikes me as a settled debate.
- The transfer of offensive AI agent to defensive utility is not that clearly described, despite its necessity. Do the authors expect that so many attacks will be discovered that an automated successful-attack-to-patch system is needed, or would there be a person in the loop? Is a person ethically required in here anywhere?
- Alternative views do not include what I suspect would be the most common alternative view, which is that these systems may fall into or be sold to the wrong hands, providing free development for cyberattackers and overall decreased security.
- Section title "Defensive Safeguarding Against Cyber Attackers: Why Not Enough" is awkward. Maybe "Defensive Safeguarding Against Cyber Attackers is Not Enough"
- Related Works is a missed opportunity to more clearly lay out the current capabilities and the rate of change of those capabilities. More discussion might result if the offense vs. defense strength and capabilities were clear.

**Support:**

3

---

> ### Author Rebuttal · Authors · 2026-03-30
>
> Thanks a lot for your valuable review and positive assessment. We would like to address your concerns as follows:
>
> > It is not clear that this is a position that needs to be argued.
>
> We appreciate this concern and would like to clarify that our primary audience is the AI community rather than the cybersecurity community. While cybersecurity practitioners have long accepted the necessity of red teaming, the AI community operates under a fundamentally different cultural and institutional framework that is currently dominated by safety alignment and the active avoidance of offensive capabilities. As we note in the introduction, most existing AI developers focus exclusively on safety-aligned AI and deliberately avoid any offensiveness [1]. The primary barrier to the development we advocate is therefore not technical but cultural and institutional, existing squarely within the AI community. We will make this target audience more explicit in the revised version to clarify why this argument needs to be made at an AI venue like ICML.
>
> > The transfer of offensive AI agent to defensive utility is not clearly described.
>
> We would like to point out that we do discuss the offense-to-defense transfer in both Section 4.2 and Section 4.3, though we agree these discussions could be more explicitly connected. In Section 4.2, we describe how cyber ranges produce detailed records of how exploits unfold that can be converted into runtime environments for high-quality trajectory collection, and how defenders gain the same capabilities to accelerate patching. In Section 4.3, we outline a concrete three-mechanism governance framework where offensive agents generate tamper-evident logs that are distilled into actionable security artifacts including automated patch suggestions and regression tests, while specialized defensive agents can be safely released to protect real systems. Regarding human oversight, we believe it is ethically required at least in the current stage of AI development, and we will add a dedicated discussion of human-in-the-loop governance in the revised version. Regarding the dangerous window between attack discovery and defense creation, our framework requires attack trajectories to remain restricted within the cyber range until defensive artifacts have been produced, and we will expand this discussion in the revised version.
>
> > Alternative views do not include the risk that these systems may fall into the wrong hands.
>
> We fully agree this represents the most credible alternative view and will add it as a substantive new section in the revised version, replacing the weaker "stalled AI progress" scenario. We will argue that the choice is not between a world where offensive AI exists only in responsible hands versus not at all, but between defenders mastering these capabilities under responsible governance versus attackers developing them first without any defensive counterpart. Avoiding offensive AI development entirely does not prevent malicious actors from developing it independently and leaves defenders permanently behind the curve.
>
> > Section title is awkward. Related Works is a missed opportunity.
>
> We will revise the section title to "Defensive Safeguarding Against Cyber Attackers is Not Enough" in the final version. Regarding Related Works, we agree and note that early 2026 represents a significant inflection point, with OpenAI releasing GPT-5.2-Codex, GPT-5.3-Codex, and Codex Security, and Anthropic releasing Claude 4.5, Claude 4.6, and Claude Code Security, all representing a qualitative shift from purely safety-aligned development. We will expand the Related Works section to document this rapid capability progression explicitly.
>
> > Timeframe, barriers, and offense-to-defense process.
>
> The transition we advocate for is already actively underway. The recent shifts by OpenAI and Anthropic [2,3] toward dual-use cybersecurity capabilities directly validate our position and suggest that cultural and institutional barriers are already beginning to break down. Beyond capability, the primary remaining barrier is cultural and institutional inertia within the AI research community and the absence of governance infrastructure for responsibly containing offensive agents. Our position paper is addressed specifically to AI researchers, organizations, and policymakers who need to build this infrastructure now. We foresee this transition accelerating significantly over the next few months, making our proposed benchmarks, trained agents, and governance frameworks immediately actionable priorities rather than speculative future directions.
>
> [1] Bengio, Y., Hinton, G., Yao, A., Song, D., Abbeel, P., Darrell, T., ... & Mindermann, S. (2024). Managing extreme AI risks amid rapid progress. Science, 384(6698), 842-845.
>
> [2] https://openai.com/index/strengthening-cyber-resilience/
>
> [3] https://www.anthropic.com/research/building-ai-cyber-defenders

---

> > ### Author Rebuttal · Reviewer_fNn4 · 2026-03-31
> >
> > Most of my concerns are relatively minor, and I believe the authors' proposed changes would improve the paper.

---

### Official Review · Reviewer_YKHC · 2026-03-07

**Significance:** 4
**Argument Clarity:** 3
**Rating:** 5
**Confidence:** 3

**Questions:**

1. Should there be further research on hybrid approaches with a human expert in the loop to direct and correct the work of AI Agents to further enhance the development and governance of security testing methods?

**Alternative Views Section:**

Yes

**Compliance With Llm Reviewing Policy A Conservative:**

Affirmed.

**Discussion Potential:**

3

**Final Justification:**

The rebuttal has addressed my main concerns and reinforced my prior positive assessment.

**Paper Summary:**

The paper states the position that in order to develop efficient cyberdefences in the era of powerful AI models the research community needs to actively work on creating AI agents able to offensively hack systems. The authors identify the assumptions of previous cybersecurity paradygms that attacking systems is resource-consuming and, therefore, limited in capacity and that given that the primary focus lies on breaking into very high-reward systems. The authors stress that given the parallelizability of AI agents and acceptance of high failure rate, a very wide range of existing systems can be attacked. The paper discusses why developing AI agents able to hack assists in preventing attacks before they happen when compared to existing defence approaches.

**Position:**

Yes

**Position In Title:**

Yes

**Related Work:**

3

**Strengths And Weaknesses:**

**Strengths**

The position stated in the paper is supported with strong reasoning, in particular, on how the emergence of accessible and modifyable AI agents capable of performing complex tasks autonomously drastically changes the existing assumptions in cybersecurity. The authors clearly state the threat model that they consider in Section 2.1 and emphasize that applying attacks at a massive scale poses a significantl problem. The paper provides a thorough overview of existing approaches to securing systems against cyber attacks such as data governance (Section 3.1), explicit constraining of model behaviour by its developers (Section 3.2) and other. It is also stated why these approaches are not sufficient given the current state of AI agent development and their availability. The authors provide ideas on how the implementation of their approach can look in reality (Table 2). Finally, the authors discuss several alternative views to their position such as challenges that may be encountered in teaching AI models to hack (Section 5.1) or limited adoption of AI Agents in Cyberattacks (Section 5.2).


**Weaknesses**

1. (Discussion potential) Some concerns come from considering the discussion potential of the stated position because approaches like red teaming and deliberately trying to break existing systems before malevolent attackers do it, are well-known in the cybersecurity community. Also, for example, the neccesity of developing strong adversarial attacks on machine learning systems is also well-discussed in the ML community. Thus, while the details of implementing and governing the AI agents able to hack, might be subject to discussion, the overall direction seems to be rather established.

2. (Formatting) It looks like Appendix (page 14) is just a pre-defined placeholder. Thus, it should be removed.
3. (Writing) Lines 226-227, right column. The sentence seems to be poorly written: "To make sure realism and coverage of attacks, we can...". Consider changing it to: "To **ensure** realism and coverage of attacks, we can.."
4. (Clarity) Consider specifying the "CTF" abbreviation for readers not familiar with cybersecurity terminology.

**Support:**

3

---

> ### Author Rebuttal · Authors · 2026-03-30
>
> Thanks a lot for your valuable review and positive assessment. We would like to address your concerns as follows:
>
>
> > Discussion potential: the overall direction seems to be rather established.
>
> We would like to provide a general clarification on the necessity and novelty of this position paper.
>
>
> **Training offensive AI agents as a necessary component of AI defense is not yet a widely accepted consensus.** We respectfully argue that while the cybersecurity community has long embraced pre-deployment red teaming and penetration testing as defensive practices, the specific position we advocate, namely that the AI community must proactively develop trained offensive AI agents as essential defensive infrastructure, is far from a consensus view within the AI research community. In fact, the dominant paradigm in AI development today remains firmly centered on safety alignment and the prevention of offensive capabilities. As we note in the introduction, most existing AI developers focus exclusively on safety-aligned AI and deliberately avoid any offensiveness [1]. This is the prevailing culture within the AI community that our position paper directly challenges, and it is a culture that is fundamentally different from the cybersecurity community's established acceptance of red teaming.
>
>
> **Current industry examples still operate at the workflow agent stage, while we also call for the security-centric training for offensive agents.** Regarding the companies cited as evidence that this position is already being implemented, we would like to draw an important distinction that is central to our argument. Companies such as XBOW, HexStrike AI, AISLE, Aikido, and Pentera are indeed developing AI-assisted penetration testing services, but as shown in Table 2 of our paper, these systems remain entirely at the workflow agent stage, where agents rely on prompting with external orchestration and fixed pipelines to find non-critical vulnerabilities and solve CTF challenges. They do not represent the trained offensive agents that we advocate for, which would use post-training from cybersecurity environments to discover zero-day vulnerabilities autonomously. This distinction is not merely technical but represents a fundamentally different and largely unexplored research direction that requires the AI community's active engagement. The workflow agent stage is mature and widely available, but the trained agent stage remains underexplored, which is precisely the gap our position paper aims to highlight and motivate. We still thank Reviewer j1aM for bringing these examples up and will include them in the paper.
>
> > Formatting: The appendix on page 14 is just a pre-defined placeholder and should be removed.
>
>
> We thank the reviewer for catching this. The placeholder appendix was included by mistake and will be removed in the final version of the paper.
>
>
> > Clarity: Consider specifying the CTF abbreviation.
>
>
> We thank the reviewer for this suggestion. We will add a definition of Capture The Flag (CTF) at its first occurrence in the paper to ensure accessibility for readers not familiar with cybersecurity terminology.
>
>
> > Should there be further research on hybrid approaches with a human expert in the loop?
>
>
> We think this is an excellent and important question that deserves serious consideration. We agree that human-in-the-loop approaches represent a promising and necessary direction, particularly in the near term before fully autonomous offensive agents can be sufficiently trusted and governed. In our governance framework in Section 4.3, we implicitly assume human oversight at several stages, including the auditing of cyber ranges, the staged release of offensive capabilities based on measured competence, and the review of attack trajectories before they are distilled into defensive artifacts. However, we agree that we do not make the role of human experts sufficiently explicit in our current framework. In the revised version, we will add a dedicated discussion of hybrid human-AI approaches, arguing that human experts serve at least three critical functions: directing agent exploration toward high-priority attack surfaces, validating discovered vulnerabilities before remediation, and providing governance oversight to ensure that offensive capabilities remain within sanctioned boundaries. We believe that such hybrid approaches are not only practically necessary in the current state of AI development but also ethically important, as they ensure meaningful human accountability over offensive AI actions. We thank the reviewer for raising this point and will incorporate it as an additional dimension of our proposed governance framework.
>
> [1] Bengio, Y., Hinton, G., Yao, A., Song, D., Abbeel, P., Darrell, T., ... & Mindermann, S. (2024). Managing extreme AI risks amid rapid progress. Science, 384(6698), 842-845.

---

> > ### Author Rebuttal · Reviewer_YKHC · 2026-04-01
> >
> > I thank the authors for addressing my concerns and for including the suggestions and improvements that were discussed in the review and in the rebuttal. I particularly appreciate the discussion on how the paper position augments existing mainstream views in the AI safety and security field.

---

### Official Review · Reviewer_rmUF · 2026-03-13

**Significance:** 4
**Argument Clarity:** 3
**Ethics Flag:** Yes
**Rating:** 5
**Confidence:** 4

**Questions:**

Why did the authors select the "stalled AI progress" scenario as a core alternative view?

**Alternative Views Section:**

Yes

**Compliance With Llm Reviewing Policy A Conservative:**

Affirmed.

**Discussion Potential:**

3

**Paper Summary:**

This position paper argues that, to defend against AI agent-driven cyber attacks, the machine learning and cybersecurity communities must proactively develop offensive AI security capabilities to predict how attacks might occur.

**Position:**

Yes

**Position In Title:**

Yes

**Related Work:**

3

**Strengths And Weaknesses:**

Strength:The paper targets a high-relevance topic for the ICML community, with a clearly articulated position and a logically consistent argumentative structure. The paper addresses a controversial, high-stakes topic with clear opposing perspectives, making it well-suited to drive meaningful debate at ICML about the responsible development of  AI security capabilities.
Weakness: The alternative view on "stalled AI progress" is a weak, low-credibility counterargument that undermines the rigor of the paper’s core position

**Support:**

3

---

> ### Author Rebuttal · Authors · 2026-03-30
>
> Thanks a lot for your valuable review and positive assessment. We would like to address your concern as follows:
>
> > Why did the authors select the "stalled AI progress" scenario as a core alternative view?
>
> We thank the reviewer for this pointed and fair criticism. We selected the "stalled AI progress" scenario as an alternative view primarily because it represents a commonly cited counterargument in broader AI risk discussions, where skeptics argue that projected risks may never materialize if capability growth plateaus. We wanted to acknowledge this perspective for completeness, as it directly challenges one of our foundational assumptions about the inevitability of increasingly capable offensive AI agents.
>
> However, we fully agree with the reviewer that this is a relatively weak counterargument that does not represent the most credible or intellectually challenging objection to our position. In hindsight, including it as a core alternative view alongside stronger counterarguments risks undermining the overall rigor of the paper, as it may give the impression that we are selecting easy-to-rebut alternatives rather than engaging seriously with the strongest objections.
>
> In the revised paper, we will demote the "stalled AI progress" scenario from a standalone section to a brief acknowledgment within a broader discussion and replace it with a stronger and more credible alternative view. Specifically, following the suggestion of Reviewer fNn4, we will add a more substantive discussion of the risk that offensive AI capabilities, once developed, may fall into or be sold to the wrong hands, providing effectively free development resources for malicious actors and potentially resulting in an overall decrease in security rather than the improvement we intend. We believe this alternative view is significantly more credible and engaging, and addressing it rigorously will strengthen rather than undermine our core position. We will also draw more explicitly on existing AI governance literature to support our rebuttal of this stronger counterargument, which additionally addresses the concern raised by Reviewer j1aM about the lack of governance-related references in Section 4.3.

---

> > ### Author Rebuttal · Reviewer_rmUF · 2026-04-01
> >
> > Thank you for your rebuttal. It addresses my concerns.  I will keep my score.

---

### Decision · Program_Chairs · 2026-04-30

**Decision:**

Accept (regular)

**Comment:**

The paper offers a highly controversial position that we should train AI agents how to hack to significantly improve security.  The key reason is that (a) AI-based hacking is inevitable (indeed, it is already happening), and (b) we need to know and understand these threats if we are to defend against them.  There are strong positions that run counter to this perspective; for example, by developing hacking agents, we are effectively developing tools that lower barriers for cyber attackers, at scale, potentially increasing overall cyber risks.  I believe that this discussion is important to have publicly, and this contribution can serve as an important part of such discussion in the AI+security community.